# Feeding Corn Silage or Grass Hay as Sole Dietary Forage Sources: Overall Mechanism of Forages Regulating Health-Promoting Fatty Acid Status in Milk of Dairy Cows

**DOI:** 10.3390/foods12020303

**Published:** 2023-01-09

**Authors:** Erdan Wang, Manqian Cha, Shuo Wang, Qianqian Wang, Yajing Wang, Shengli Li, Wei Wang

**Affiliations:** 1State Key Laboratory of Animal Nutrition, Beijing Engineering Technology Research Center of Raw Milk Quality and Safety Control, College of Animal Science and Technology, China Agricultural University, Beijing 100193, China; 2Key Laboratory of Feed Biotechnology of the Ministry of Agriculture and Rural Affairs, Institute of Feed Research, Chinese Academy of Agricultural Sciences, Beijing 100081, China

**Keywords:** forage, lipid metabolism, gastrointestinal tract, mammary glands, milk, health-promoting fatty acids

## Abstract

Different dietary forage sources regulate health-promoting fatty acids (HPFAs), such as conjugated linoleic acids (CLAs) and omega-3 polyunsaturated fatty acids (n-3 PUFAs), in the milk of lactating cows. However, the overall mechanism of forages regulating lipid metabolism from the gastrointestinal tract to the mammary glands (MGs) is not clear. Three isocaloric diets that contained (1) 46% corn silage (CS), (2) a mixture of 23% corn silage and 14% grass hays (MIX), and (3) 28% grass hays (GH) as the forage sources and six cannulated (rumen, proximal duodenum, and terminal ileum) lactating cows were assigned to a double 3 × 3 Latin square design. Our results show that a higher proportion of grass hay in the diets increased the relative contents of short-chain fatty acids (SCFAs), CLAs, and n-3 PUFAs. The lower relative content of SCFA in the milk of CS was predominantly due to the reduction in acetate production in the rumen and arteriovenous differences in the MG, indicating that the de novo synthesis pathways were inhibited. The elevated relative contents of total CLA and n-3 PUFA in the milk of GH were attributed to the increases in apparent intestinal digestion and arteriovenous differences in total CLA and n-3 PUFA, together with the higher Δ^9^-desaturase activity in the MG. In conclusion, this study provides an overall mechanism of dietary forages regulating HPFA status in the milk of dairy cows.

## 1. Introduction

Milk and dairy products are ubiquitous nutritive foods that are frequently included as important components of a healthy and balanced diet. Fatty acid composition plays an important role in milk quality, and substantially affects long-term human health [1,2,3]. Previous studies on human subjects have confirmed the role of health-promoting fatty acids (HPFAs), such as conjugated linoleic acids (CLAs) [4], omega-3 polyunsaturated fatty acids (n-3 PUFAs) [5], and short-chain fatty acids (SCFAs; carbon chain ≤ 6;) [6], in preventing several chronic diseases such as cardiovascular diseases, some forms of cancer, obesity, and diabetes [2,3]. Particularly, some HPFAs, such as CLA and butyric acid (C4:0), in the human diet are mainly derived from or relatively rich in the meat and milk of ruminants [4,7]. However, a relatively high proportion of saturated fatty acids (SFAs), such as dairy-origin hypercholesterolemic fatty acids (HFAs; C12:0 + C14:0 + C16:0), raises some concerns about dairy consumption [7].

The fatty acid composition of milk in lactating cows depends mainly on lipid metabolism in the gastrointestinal tract (GIT) and mammary glands (MGs) [8,9,10]. Volatile fatty acids (VFAs) from rumen fermentation, such as acetate (60–70% of all VFA) and butyrate-derived β-hydroxybutyrate, change the de novo synthesis in the MGs [11]. The hydrolysis and biohydrogenation of lipids via microbial activity alters the SFAs, unsaturated fatty acids (UFAs), and CLAs in the rumen [12,13]. The low pH [14] and high concentration of bile acids [10], as well as the passage rate (Kp) [10,15] of the digesta, influence the solubility and digestibility of fatty acids (FAs) in the small intestine. Furthermore, desaturase activity in the MG regulates the endogenous production of monounsaturated fatty acids (MUFAs) and nearly all CLAs [16]. Moreover, even though most digestion of FAs occurs in the jejunum, biohydrogenation also occurs in the large intestine [17]. Therefore, the determination of the duodenal and ileal flows of FA is necessary to assess the digestibility of individual or total FAs in ruminants accurately. Nevertheless, only a few studies have assessed FA utilization by rumen microbiota and small intestinal digestion of FA in dairy cows using duodenal and ileal cannulated cows [18]. Although these key processes affecting HPFA content in milk have been widely studied, most of the previous studies have focused on a single process or the nutritional aspect, which lacks a comprehensive whole-system modeling approach to highlight the overall biology of the milk production process [10,19].

Corn silage, alfalfa hay, and oat hay are the main dietary forage sources for intensive dairy farms all over the world, and contribute to the global health and adequacy of milk and dairy products for the increasingly growing human population. Forage is important for ruminants; not only it is a source of metabolic energy, but it also contains sufficient physically effective neutral detergent fiber (peNDF) to stimulate rumination and saliva production, which buffers the rumen, thereby promoting rumen health [20]. Furthermore, dietary forage sources have a particularly strong effect on the dietary indigestible neutral detergent fiber (iNDF) content, which is an important parameter in mechanistic rumen models [20,21,22,23,24]. High-producing dairy cows generally receive up to 5–6% of fat in their diets, with approximately 3% coming from the forages and grains, and the rest is added as supplemental fat [10]. Fat supplements such as palm oil have been widely used; however, dietary supplementation of n-3-PUFA-rich feeds does not easily increase quantities of milk fat [25]. Although the FA contents in forages are low (1–3%), the UFA proportions are usually high (more than 50%) [26], resulting in the forages being one of the most economical UFA sources in dairy diets. Increasing the proportion of corn silage in the diets increased the economic effectiveness, but decreased the relative contents of HPFA, such as CLA and n-3 PUFA, in raw milk [1,19,27] and cheese [28]. Furthermore, prediction models have been developed to discriminate between “grass hay milk” and “corn silage milk” based on their FA profiles [19,29]. However, the elevated concentrations of CLA and n-3 PUFA are attributed to incomplete hydrogenation in the rumen or the high α-linolenic acid (C18:3 n-3) content of forage grasses, which is debated [19,30]. Additionally, the claims of de novo synthesis of FA or SCFA contents in milk by increasing the proportion of corn silage as dominant forage (33–35% of diet DM) are highly contested [19,31]. The lack of a comprehensive understanding of the overall regulatory mechanism of FA metabolism in the GIT and MG prevents more achievable outcomes.

Therefore, the objectives of the present study were to investigate the effects of different dietary forage sources on 3 systems/organs: (1) the dietary iNDF content and FA composition, (2) VFA concentration in the rumen, FA biohydrogenation and digestion in the GIT, as well as (3) the arteriovenous difference and desaturase activity in the MGs. Our findings can be used to gain insight into the holistic landscape of lipid metabolism from the GIT to the MGs, and the overall mechanism of forages regulating HPFA status in milk of dairy cows.

## 2. Materials and Methods

The experiments were performed in accordance with the Guide for the Care and Use of Agricultural Animals in Agricultural Research and Teaching (Protocol number: 2013-5-LZ). All experimental procedures, including ruminal and intestinal cannulation, were approved by the Institutional Animal Care and Use Committee of the China Agricultural University (Permit number: AW61110202-2; Date: 25 August 2019).

### 2.1. Animals, Diets, and Experimental Design

Six second-parity lactating Holstein cows previously fitted with cannulas in the rumen (10 cm internal diameter ruminal cannulas, Anscitech Farming Technology Co., Ltd., Wuhan, China), proximal duodenum, and terminal ileum (T-shaped intestinal cannulas, Anscitech Farming Technology Co., Ltd., Wuhan, China) were assigned to a double 3 × 3 Latin square design with three dietary treatments and three periods. Each period was 28 d, consisting of 14 d of adaptation to diet followed by 14 d of sampling and data collection. The cannulas were fitted 2 months before the start of the experiment. The duodenal cannula was placed approximately 10 cm distal to the pylorus and proximal to the common bile and pancreatic duct. The ileal cannula was placed approximately 10 cm proximal to the cecum. At the beginning of the experiment, the days in milk (DIM), dry matter intake (DMI), daily milk yield (MY), and body weight (BW) (mean ± standard deviation) of the cows were found to be 101.67 ± 6.35 d, 23.68 ± 0.88 kg/d, 31.43 ± 4.59 kg/d, and 623 ± 25 kg, respectively. Cows were housed in an individual pen equipped with a roughage intake control system (RFID, Zhenghong Company, Shanghai, China). The pen was bedded with rice hull that was cleaned every 3 d by removing all bedding and renewing it with fresh rice hull. Cows were milked thrice daily (at 0630, 1330, and 2030 h) using a DeLaval milking system (9JGJ-2×12 herringbone milking system, DeLaval (Tianjin) Co., Ltd., Tianjin, China). The cows were given free access to water during the experiment.

Three isocaloric diets (Table 1) were formulated to meet or exceed net energy of lactation requirements for cows of 650 kg of BW producing 30 kg/d of milk [32] with different forage sources (DM basis): (1) 46% corn silage as the sole dietary forage source (CS), (2) a mixture of 23% corn silage and 14% grass hays (6% alfalfa hay and 8% oat hay) as the dietary forage source (MIX), and (3) 28% grass hays (12% alfalfa hay and 16% oat hay) as the sole forage source (GH). The diets were administered twice daily at 0700 and 1900 h in amounts that allowed 10% orts. The corn silage was harvested at a stubble height of 40 cm, theoretical chop length of 2.1 cm, and roll clearance of 1.5 mm. The chemical composition and FA profile of the varied ingredients are presented in Table 2. 

### 2.2. Sampling and Measurements

The feed intake of each cow was monitored daily using a roughage intake control system (RFID, Zhenghong Company, Shanghai, China) during the experimental phase, and the DMI of each cow was calculated based on the DM contents of diets offered and refused (orts), as described in our previous study [33]. Furthermore, diets and orts were collected and combined into one sample per period and stored at −20 °C for chemical and FA analysis. Milk production was recorded for the first five consecutive days (d 15–19) of each sampling period, and the milk sample was collected on d 18–19 using milk sampling devices (DeLaval (Tianjin) Co., Ltd., Tianjin, China). One 50 mL aliquot of each composited milk sample (4:3:3; morning, afternoon, and evening) was mixed with potassium dichromate and stored at 4 °C until it was sent to the Dairy Herd Improvement Center (Beijing, China) for the determination of milk composition. Milk fat (%), protein (%), and lactose (%) were analyzed using an automated near-infrared milk analyzer (Seris300 CombiFoss FT+; Foss Electric, Hillerød, Denmark). Another 50 mL aliquot of milk was stored at −80 °C for later analysis of the FA profile. Average daily yields of protein, fat, and lactose were calculated from these data for each period. Yields of fat-corrected milk (3.5%) and energy-corrected milk were calculated according to National Research Council (NRC) equations.

The iNDF_288_ contents of the diets and varied forage ingredients were determined by ruminal incubation for 288 h; the NDF and iNDF_288_ contents of forages have been reported in our previous study [34]. The in situ incubation procedure followed a proposal for a standardized method for forage ingredients [32,35]. Briefly, approximately 5 g of samples were filled into heat-sealed nylon bags in six repetitions. The nylon bags were incubated in the rumen for 288 h to determine iNDF_288_ contents.

Ruminal, duodenal, ileal, and rectal digesta samples were taken every 3 h during d 15–19 of each sampling period in a staggered schedule with a total of eight samples per site. Approximately 250 g of spot digesta samples were collected at each time point from each cow. The pH of the spot sample was measured immediately using a handheld pH meter (Starter 300; Ohaus Instruments Co. Ltd., Shanghai, China). At the end of each period, spot samples were pooled based on each cow within the period to provide a composite sample of digesta. Subsequently, the pooled samples were aliquoted and stored at −80 °C until FA and bile acid profile analysis.

Blood samples were collected from the coccygeal artery (arterial plasma sample, A) and mammary veins (mammary venous plasma sample, V) at 0600 h prior to the morning feeding and milking during d 18–19 of each sampling period. The cows were standing for at least 10 min prior to blood sampling. Arterial blood is considered to be sufficiently mixed, and it can be obtained from any source. Thus, the composition of arterial blood from the tail was assumed to be representative of that of the mammary arterial supply [36]. The plasmas were obtained by centrifugation (L420-A centrifuge, Cence Group, Hunan, China) at 3000× *g* for 15 min at 4 °C. Subsequently, the plasmas were aliquoted (1 mL) into appropriately labeled cryovial tubes and frozen at −80 °C until further analysis.

### 2.3. Estimation of the Digesta Flow across the Gastrointestinal Tract (GIT)

The rumen pool, digesta flow across the GIT, and Kp through the rumen were estimated following the modified methodology described by Pantoja J. et al. [37] during the last 9 days (d 20–28) of each experimental period. Briefly, polyethylene glycol (PEG) mixture (50 g PEG + 1 L warm water) and chromic oxide (Cr_2_O_3_) gelatin capsules (5.0 g per dose) were administered via the ruminal cannulas twice daily (0700 and 1900 h) from d 20 to 28. Duodenal, ileal, and rectal samples were taken every 3 h during the last 4 days (d 25–28) in a staggered schedule that allowed one sample for each 4 h of a 24 h period with a total of eight samples per site. Ruminal digesta were evacuated and measured on d 28 of each experimental period. The flow of digesta was intermittent, gushes varied between 100–500 mL, and was separated by intervals of less than 1 to more than 15 min. Approximately 250 g of spot digesta or feed samples were collected at each time point from each cow. At the end of each period, spot samples were pooled based on each cow within the period to provide a composite sample of digesta. Subsequently, the spot and pooled samples were stored at −20 °C until further chemical and marker analysis. Digesta flow entering the proximal duodenum and terminal ileum and fecal output were estimated as the means of two markers (PEG and Cr_2_O_3_) [38]. The DM flow of each marker were calculated by dividing the concentration of marker dosage given to the cows by that of the marker measured in duodenal, ileal, and rectal samples [39]. The Kp through the rumen was determined using the following equation [40]:(1)$$\mathrm{Passage}\;\mathrm{rate}\;(\%/h)=\frac{\mathrm{Duodenal}\;\mathrm{flow}\;\mathrm{of}\;\mathrm{component}\;(\mathrm{kg}/d)}{\mathrm{Ruminal}\;\mathrm{pool}\;\mathrm{of}\;\mathrm{component}\;(\mathrm{kg})}\;\times \;\frac{100\%}{24\;h/d}$$

### 2.4. Analytical Procedures

Feedstuff, diets, orts, and pooled digesta samples were thawed at room temperature and analyzed for dry matter (method 950.15), starch (method 996.11), nitrogen (method 984.13), ether extract (method 920.39), and ash (method 924.05) following the methods described by the Association of Official Analytical Chemists (AOAC). The contents of neutral detergent fiber (NDF) and acid detergent fiber (ADF) were analyzed using a fiber analyzer (A2000i; Ankom Technology, Fairport, NY, USA). Concentration of NE_L_ of the diets was calculated according to NRC (2001) [32]. The molar concentrations of VFA in ruminal and duodenal spot samples (collected at 0600 h prior to the morning feeding and milking) were determined as described in our previous study [41]. A gas chromatograph (6890N; Agilent technologies, Avondale, PA, USA) equipped with a capillary column (HP-INNOWax 19091N-213, Agilent) was used. All the above analyses for all samples were conducted in triplicate.

All FA profile analyses were conducted by the Ministry of Agriculture Feed Industry Centre (MAFIC, Beijing, China) in duplicate. The FA methyl esters (FAMEs) for feedstuff, diets, and pooled digesta samples were prepared according to the method of Sukhija and Palmquist [42], and those for plasma and milk samples were prepared following the Tsiplakou method [43]. The analyses of FAMEs are elaborated in the National Standard (GB 5413.27-2010) with slight modifications. Undecanoic acid (C11:0) was added to the sample as the internal standard, followed by adding 4.0 mL of 5% chloroacetyl methanol solution and 1 mL of hexane to the sample. The sample was incubated in an 80 °C water bath for 2 h after vortex. After taking out the sample from the water bath and cooling it for 20 min, 5.0 mL of 7% potassium carbonate was added to the sample. The sample was centrifuged at 800× *g* at 25 °C for 5 min. The organic layer was filtrated (0.2 μm filter membrane) and transferred to a gas chromatography autosampler vial. FA profile was analyzed using an Agilent 6890 gas chromatograph equipped with a DB-23 capillary column (60 m × 250 µm × 0.25 μm, Agilent Technologies Co., LTD, Beijing, China) and flame ionization detector with helium as the carrier gas (2.0 mL/min). The individual and total FA contents were obtained according to the internal standard content. Intestinal apparent digestibility was then calculated based on digesta flow and FA concentration in the duodenal and ileal digesta samples.

The arterial and venous plasma glucose and acetate were determined by an automatic biochemical analyzer (CLS880, ZECEN Biotech Co., Ltd., Weifang, China) using commercially available kits (ZECEN Biotech Co., Ltd., Weifang, China) according to their instructions. Nonesterified fatty acid (NEFA), β-hydroxybutyrate (BHB), and very-low-density lipoprotein (VLDL) concentrations were detected using an enzyme-labeled instrument (Thermo Multiskan Ascent, American) with enzyme-linked immunosorbent assay (ELISA). The arteriovenous (AV) differences across the MG were calculated following the description of Hanigan et al. [44]: AV difference = arterial plasma − mammary venous plasma. Positive AV differences indicate metabolite removal from plasma, whereas negative values indicate net metabolite release from the MG.

Bile acid profiles in duodenal and ileal digesta were quantified according to the LC-MS/MS method of García-Cañaveras et al. [45]. Cholic acid (CA), α-muricholic acid (α-MCA), β-muricholic acid (β-MCA), ursodeoxycholic acid (UDCA), tauro-α-muricholic acid (Tα-MCA), taurohyocholic acid (THCA), taurohyodeoxycholi acid (THDCA), glycochenodeoxycholic acid (GCDCA), glycohyocholic acid (GHCA), Deoxycholic acid (DCA), tauro-ω-muricholic acid (Tω-MCA), glycocholic acid (GCA), glycolithocholic acid (GLCA), glycohyodeoxycholic acid (GHDCA), and deuterated internal standards lithocholic acid-D4 (LCA-D4), deoxycholic acid-D4 (DCA-D4), cholic acid-D4 (CA-D4), and glycochenodeoxycholic acid-D4 (GCDCA-D4) were purchased from Steraloids Inc (Newport, RI, USA). Lithocholic acid (LCA), ω-muricholic acid (ω-MCA), chenodeoxycholic acid (CDCA), hyocholic acid (HCA), hyodeoxycholic acid (HDCA), taurocholic acid (TCA), taurolithocholic acid (TLCA), tauroursodeoxycholic acid (TUDCA), taurodeoxycholic acid (TDCA), tauro-β-muricholic acid (Tβ-MCA), taurochenodeoxycholic acid (TCDCA), glycodeoxycholic acid (GDCA), glycoursodeoxycholic acid (GUDCA), glyco-β-muricholic acid (Gβ-MCA), and deuterated internal standard taurocholic acid-D4 (TCA-D4) were purchased from Cayman Chemical Inc (Michigan, USA). Deuterated internal standards taurolithocholic acid-D5 (TLCA-D5) and tauroursodeoxycholic acid-D5 (TUDCA-D5) were purchased from Toronto Research Chemicals Inc (Toronto, ON, Canada). Pooled digesta samples (approximately 100 mg) were placed in 2 mL tubes containing 600 µL of cold methanol and 200 µL of internal standards stock solution (1 μg/mL). Then, digesta samples were homogenized twice for 25 s at 10,000 rpm at 4 °C. The tubes were centrifuged at 3000× *g* for 5 min at 4 °C, and the supernatants were transferred to new tubes. A second bile acid extraction was performed, and the two extraction supernatants were pooled and aliquoted (150 µL). The aliquots were evaporated to dryness and reconstituted in 100 µL of 50% methanol solution, centrifuged at 10,000× *g* for 10 min at 4 °C, and transferred into chromatography autosampler vials. Bile acids were quantified using an Agilent 1200 ultra-high phase liquid chromatography system and 6460 triple quadrupole mass spectrometer (Agilent Technologies Co., LTD, Beijing, China). The separation was performed on a Waters BEH C18 ultra-high-performance liquid chromatography column (1.7 μm, 2.1 × 100 mm) at 65 °C. Gradient elution was performed with 0.1% formic acid and 0.1% formic acetonitrile as mobile phases at the flow rate of 0.5 mL/min. Mass spectrometry was performed in negative ion mode.

### 2.5. Statistical Analysis

All data were analyzed using the Proc Mixed procedure of SAS (version 9.4; SAS Institute Inc., Cary, NC). Before analysis, the data were tested for normal distribution using the PROC UNIVARIATE procedure. Data points with studentized residuals greater than ±3.0 were considered outliers and excluded from the analysis. The best variance and covariance structure models were selected based on the values of Akaike and Bayesian information criterion. The variables for DMI, milk production, fermentation characteristics, digesta flow, intestinal digestion, plasma metabolites, and milk FA composition were analyzed as a duplicated 3 × 3 Latin square design using the following model:(2)$$Y_{\mathrm{ijkl}}{=\mu +T}_{i}{+P}_{j}{+S}_{k}{+C}_{l(k)}{+e}_{\mathrm{ijkl}}$$
where *Y_ijkl_* = the dependent variable, *μ* = the overall mean, *T_i_* = the fixed effect of the *i*th diet treatments (*i* = 1 to 3), *P_j_* = the fixed effect of the *j*th period (*j* = 1 to 3), S*_k_* = the random effect of the *k*th square (*k* = 1 to 2), C*_l(k)_* = the random effect of the *l*th cow within the *k*th square (*l* = 1 to 3), and *e_ijkl_* = residual error, assumed to be normally, identically, and independently distributed (NIID). Orthogonal polynomial contrasts were used to determine the linear and quadratic effects of corn silage levels in the diets. Least square means were reported throughout. Differences were considered significant at *p* < 0.05 and tendencies at 0.05 ≤ *p* < 0.10.

## 3. Results

### 3.1. Diets, DMI, and Lactation Performance

The relative contents of C18:3 n-3 and SFA were the lowest in CS, and the highest in GH, with MIX being intermediate (Table 1). Meanwhile, the relative contents of C18:2 n-6c and PUFA were the highest in CS, and the lowest in GH, with MIX being intermediate. The iNDF_288_ contents (% of DM) of CS, MIX, and GH were 8.07%, 8.36%, and 8.34%, respectively. A linear increase (*p* = 0.03) in DMI was observed as the proportion of corn silage in the diets increased, with the values of 24.11, 23.09, and 22.24 kg/d for CS, MIX, and GH, respectively (Table 3). However, no significant effects of forage (*p* > 0.10) on milk production and milk composition were observed, despite the increase in DMI as the proportion of corn silage increased in the diets. Furthermore, increasing the proportion of corn silage in the diets tended to linearly decrease the feed efficiency (*p* = 0.08).

### 3.2. Fermentation Characteristics in the Rumen and Duodenum

The ruminal and duodenal fermentation characteristics are presented in Table 4. No significant differences were observed in ruminal pH or molar concentration of total VFAs among the three diets (*p* > 0.10); in contrast, quadratic decreases in acetate (*p* = 0.04) and A:P ratio (*p* = 0.02) were observed as the proportion of corn silage in the diets increased. pH and total VFA concentration in the duodenum tended to show a linear increase (*p* = 0.07) and quadratic decrease (*p* = 0.08), respectively, as the proportion of corn silage increased in the diets. The R–D difference in acetate was highest in MIX and lowest in CS, with a significant quadratic effect of the dietary corn silage level (*p* = 0.03).

### 3.3. Fatty Acids Flow across Gastrointestinal Tract (GIT) and Kp through the Rumen

Significantly linear increases in DMI (*p* = 0.03), and digesta DM flow entering the duodenum (*p* = 0.01) and leaving the ileum (*p* < 0.01) and rectum (*p* < 0.01), were observed as the proportion of corn silage in the diets increased (Figure 1a). Accordingly, significant linear increases in total FA intake (*p* = 0.04), total FA flow entering the duodenum (*p* < 0.01), and leaving the ileum (*p* < 0.01) and rectum (*p* < 0.01) were observed as the proportion of corn silage in the diets increased (Figure 1b). Additionally, a significant linear increase in Kp through the rumen (*p* = 0.01) was detected as the proportion of corn silage in the diets increased (Figure 1c).

The dominant and HPFA flows across the GIT are shown in Appendix A. Higher dietary corn silage proportion linearly decreased C18:3 n-3 intake (*p* = 0.04) and duodenal flow (*p* = 0.02), whereas it tended to increase C18:3 n-3 ileal flow (*p* = 0.06) and significantly increased C18:3 n-3 fecal flow (*p* = 0.03). Additionally, there was a linear decrease in total CLA flow entering the duodenum (*p* = 0.01), and an insignificant difference in digesta flow leaving the ileum (*p* > 0.10). Contrary to its effects on C18:3 n-3, the higher proportion of corn silage linearly increased OCFA duodenal flow (*p* = 0.02) and ileal flow (*p* = 0.02). However, dietary forage sources had no significant effects (*p* > 0.10) on the ruminal biohydrogenation of unsaturated C18 FA (Appendix A). Interestingly, the relative concentrations of C18:3 n-3 tended to decrease in duodenal digesta (*p* = 0.06; Appendix A), with no significant differences in the ileal digesta (*p* > 0.10; Appendix A), as the proportion of corn silage in the diets increased. Meanwhile, the higher dietary corn silage proportion linearly decreased the relative concentrations of total CLA in duodenal (*p* = 0.02) and ileal digesta (*p* = 0.01).

### 3.4. Apparent Intestinal Digestion of Dominant and Health-Promoting Fatty Acids (HPFAs)

The apparent intestinal digestion (duodenal FA flow–ileal FA flow) of dominant and HPFAs are shown in Figure 2. Higher dietary corn silage proportion tended to linearly increase the apparent intestinal digestion of total FA (*p* = 0.06; Figure 2a) and OCFA (*p* = 0.07; Figure 2h), and linearly increased the digestion of C18:0 (*p* = 0.02; Figure 2c) and C18:1 n-9c (*p* = 0.03; Figure 2d). In contrast, higher dietary corn silage proportion linearly decreased the apparent digestion of C16:0 (*p* = 0.02; Figure 2b), total CLA (*p* = 0.04; Figure 2f), and n-3 PUFA (*p* < 0.01; Figure 2g).

Furthermore, higher dietary corn silage proportion tended to decrease, linearly in some cases, the apparent digestibility of total FA (*p* = 0.03), C16:0 (*p* = 0.01), C18:0 (*p* = 0.07), C18:3 n-3 (*p* = 0.01), MCFA (*p* = 0.03), LCFA (*p* = 0.03), SFA (*p* = 0.01), HFA (*p* = 0.01), and n-3 PUFA (*p* = 0.01) in the small intestine (Appendix A).

### 3.5. Bile Acid Profile in Duodenal and Ileal Digesta

As the proportion of corn silage in the diets increased, the content of total bile acids in duodenal digesta tended to decrease (*p* = 0.06) with the values of 12.27 × 10^7^, 13.53 × 10^7^, and 10.61 × 10^7^ ng/g, respectively (Table 5). The contents of total bile acids in ileal digesta decreased sharply compared to those in duodenal digesta, in which they tended to increase (*p* = 0.08) to the values of 4.79 × 10^5^, 5.90 × 10^5^, and 7.22 × 10^5^ ng/g, respectively.

The TCA and GCA were the dominant bile acids in the duodenal digesta of the dairy cows. Higher proportion of corn silage in the diets tended to decrease (*p* = 0.09) the TCA content in duodenal digesta to the values of 8.29 × 10^7^, 8.40 × 10^7^, and 7.34 × 10^7^ ng/g, respectively. Additionally, the GCA content in duodenal digesta decreased to the values of 1.80 × 10^7^, 1.75 × 10^7^, and 1.48 × 10^7^ ng/g, but there was no statistical difference (*p* > 0.10). Intriguingly, higher dietary corn silage proportion linearly decreased the contents of LCA (*p* = 0.04), CDCA (*p* = 0.02), DCA (*p* = 0.03), β-MCA (*p* < 0.01), and CA (*p* < 0.01) in duodenal digesta, and quadratically decreased the contents of HDCA (*p* < 0.01), GLCA (*p* < 0.01), GDCA (*p* = 0.04), and TCDCA (*p* = 0.02) in duodenal digesta.

### 3.6. Arteriovenous (AV) Difference

As expected, linearly lower AV differences in C18:3 n-3 (*p* = 0.04), total CLA (*p* = 0.04), n-3 PUFA (*p* = 0.04), and acetate (*p* = 0.01) were observed when dietary corn silage proportion increased (Table 6). Meanwhile, tendencies of quadratically increasing the AV differences in VLDL (*p* = 0.07) and LCFA (*p* = 0.09) were observed. However, significantly higher AV differences in MUFA (*p* = 0.03) were observed in MIX and GH compared to that of CS. Notably, a linearly lower AV difference in acetate (*p* = 0.01) was observed when dietary corn silage proportion increased, whereas there were no significant differences (*p* > 0.10) in metabolites of BHBA, glucose, and NEFA. The FA composition and other metabolites in arterial and venous plasma of cows fed diets with different forage sources are presented in Appendix A, respectively.

### 3.7. Fatty Acid Composition in Milk, Synthesized Origins, and Desaturase Activity

Most of the FA underwent important modifications as the proportion of corn silage in the diets increased (Appendix A). Higher proportion of corn silage in the diets linearly decreased the relative contents of SCFA (*p* = 0.04) and MCFA (*p* = 0.07), at the expense of LCFA (*p* = 0.06) content, to such an extent that the CS diet had the highest levels of LCFA and lowest levels of SCFA and MCFA (Figure 3a).

Additionally, the relative content of PUFA linearly decreased (*p* = 0.04) with no significant differences (*p* > 0.10) in SFA and MUFA. The substantial increases (*p* < 0.05) in HPFA were those of butyric acid (7.7%; Figure 3c), CLA-c9t11 (31.3%), total CLA (25.9%; Figure 3d), C18:3 n-3 (31.3%), and n-3 PUFA (27.7%; Figure 3e) in GH compared to those in CS. Consequently, a linear rise (*p* = 0.02) was observed in the n-6/n-3 ratio, with values of 8.90, 8.17, and 7.68 for CS, MIX, and GH, respectively.

Significantly higher (*p* = 0.02) de novo synthesized FA origins in MIX and GH diets compared to CS were observed (Table 7). Meanwhile, higher levels of preformed synthesized FA were observed in CS compared to MIX and GH (*p* = 0.06). Additionally, Δ^9^-desaturase activities in C14:1/(C14:0 + C14:1) (*p* = 0.06), C16:1/(C16:0 + C16:1) (*p* = 0.03), and C18:1/(C18:0 +C18:1) (*p* = 0.08) tended to linearly decrease as the proportion of corn silage in the diets increased.

## 4. Discussion

Corn silage contained lower levels of C18:3 n-3 and iNDF_288_ compared with grass hay, which was driven to lower contents of C18:3 n-3 and iNDF_288_ in the diets with higher proportion of corn silage. Higher dietary corn silage proportion linearly or quadratically decreased the acetate concentration in the rumen and arteriovenous differences in acetate in the MGs, indicating that the de novo synthesis pathways were inhibited. Increasing dietary corn silage proportion linearly increased DMI, total FA flow across the GIT, and duodenal pH, but decreased the concentrations of total and dominant bile acids in the duodenum, resulting in lower digestibility but higher intestinal digestion of LCFA. Furthermore, higher arteriovenous differences in VLDL and LCFA, together with higher content of LCFA in milk, indicate that the higher proportion of corn silage stimulated the preformed synthesis pathways. Conversely, increasing the proportion of grass hay in the diets increased intestinal digestion and arteriovenous differences in n-3 PUFA and total CLA, together with higher Δ^9^-desaturase activity in the MG, which resulted in higher contents of n-3 PUFA and total CLA in the milk. Taken together, increasing the proportion of grass hay in the diets increased the relative contents of HPFA in the milk, specifically, SCFA, CLA, and n-3 PUFA. The overall mechanism of different dietary forage sources regulating the HPFA in milk of lactating cows is summarized below based on the present results (Figure 4).

### 4.1. DMI and Lactation Performance

The lactation performance of the cows was closely linked to DMI and nutrient digestibility, although the potential performance of the cows fed corn silage might have been strategically limited to enhance milk quality. In agreement with the present study, a higher DMI and insignificant changes in milk production and composition after replacing corn silage with alfalfa hay as the sole forage source were previously reported [46]. Furthermore, there were insignificant changes in DMI, MY, and milk composition after replacing maize silage with grass/lucerne hay [19]. After replacing corn silage with alfalfa silage, contentious results on DMI, MY, and milk composition were reported [47]. The differences in results could be because of different NDF and iNDF concentrations among the diets. Corn silage quality, such as peNDF and iNDF content, fermentation parameters, and mycotoxin content are the limiting factors for high proportion utilization of corn silage in dairy diets. Our previous study reported that corn silage contained lower levels of iNDF_288_ compared with alfalfa and oat hay [34], which led to a lower iNDF_288_ content in corn-silage-based diets and mainly attributed to higher DMI and Kp [20,47]. Although the NDF concentration was kept constant, increasing the proportion of alfalfa silage in the diets increased iNDF concentration [47]. The values of Kp in the current study are within the range reported in a meta-analysis [48] that was conducted to study the effect of forage type on particle passage rate, with iNDF as an internal marker based on the NRC (average of 5.01%/h) [32] and Cornell Net Carbohydrate and Protein System (CNCPS, average of 4.24%/h) models [49]. Furthermore, decreased nutrient digestibility was reported to be in agreement with increased Kp and decreased mean ruminal retention time [15,47]. Consequently, the higher DMI together with lower nutrient digestibility in the cows fed CS contributed to similar lactation performance among the three diets.

### 4.2. Ruminal Fermentation Characteristics and De Novo Synthesis Pathways

Acetate is the most abundant VFA (60–70% of total VFA); it is produced by the fibrolytic microbiotic activity in rumen fermentation, and contributes approximately 55% of the net energy absorbed as SCFA [50]. Among VFA, acetate and butyrate-derived β-hydroxybutyrate are the major precursors for de novo synthesis of FA, with a minor role of propionate in the synthesis of OCFA, such as C13:0, C15:0, and C17:0 [51], which contribute approximately one-half of the total milk fat origins in the MGs [52]. Furthermore, increasing ruminal acetate absorption would modify gene expression and cellular physiology, and therefore increase milk fat by increasing de novo FA synthesis [11].

In the present study, the total VFA concentration was not affected by diet; however, quadratic decreases in acetate and the A:P ratio were observed as the proportion of corn silage in the diets increased. Lopes et al. (2015) [47] reported the same tendencies when feeding diets with different proportions of corn silage to alfalfa silage to cows. Several previous studies [46,53,54] also reported reduced ruminal acetate concentration when alfalfa silage was partially replaced by corn silage, or when comparing corn-silage-based diets with alfalfa-hay- or silage-based diets. The highest acetate concentrations and A:P ratio were observed in MIX compared with those of CS and GH; this result can be explained by the higher potential digestibility of NDF because of the combined effect of roughages [37]. Furthermore, we observed lower ruminal absorption and MG uptake of acetate in cows fed diets with a higher proportion of corn silage, and these factors were highest in cows fed the MIX diet, with a significant quadratic effect of dietary corn silage levels. Therefore, higher concentrations of de novo synthesized FA, SCFA, and especially butyric acid were observed in the milk of cows fed diets with a higher proportion of grass hay. Taken together, higher dietary grass hay proportion increased SCFA (especially butyric acid) in milk predominantly through increases in ruminal acetate production and uptake by MG, indicating that the de novo synthesis pathways were stimulated.

However, some previous studies increased the dietary corn silage proportion as a dominant forage source, causing higher concentrations of de novo synthesis of FA and SCFA, which are inconsistent with our results [19,55]. The differences could be because of the experimental diets, which were neither isocaloric or isonitrogenous, nor a partial replacement of corn silage with grass hay. Acetate deficiency theory was proposed as a mechanism of classic diet-induced milk fat depression (MFD) after feeding cows highly fermentable diets that increase propionate production [56]. Furthermore, biohydrogenation-induced MFD is another common example of dietary regulation of de novo synthesized FA in the milk of dairy cows [10]. Moreover, adequate fiber was provided in all these studies; thus, it was a low-risk diet for biohydrogenation-induced MFD, with low concentrations of alternate biohydrogenation intermediates (C18:1 t10 and CLA-t10c12) in milk.

### 4.3. Dietary Fatty Acid Digestion and Preformed Synthesis Pathways

Preformed LCFAs, which provide the other half of total milk fat origins in the MG, mostly come from diet intake [52,57]. Diet composition and intake (mainly LCFA), quantities of biohydrogenation intermediates produced and absorbed in the GIT, transportation in plasma, and uptake and metabolism by the MG play a significant role in regulating the dominant and HPFA in milk by preformed synthesis pathways [9,10,18].

Notably, different forages contain distinct lipid contents and FA compositions [26]. Compared with corn silage, alfalfa and oat hay contain higher levels of C16:0 and C18:3 n-3 and lower levels of C18:2 n-6c, which account for the similar FA compositions in the diets of the present study. In agreement with the meta-analysis [18], higher DMI resulted in higher total FA intake and total FA flow crossing the GIT, despite the formulation of three experimental diets with the same level of FA. Previous studies have preferred to compare the apparent FA digestibility in the small intestine [18]. However, lower FA intake resulted in lower duodenal FA flow, potentially leading to higher apparent FA digestibility and lower FA digestion in the small intestine [58]. Our apparent FA digestibility data are consistent with those of previous studies [18]. The digestibility of saturated FA decreases with increasing chain length, and unsaturation increases the digestibility of the dominant FA. The digestibilities of the total FAs in the current study are within the ranges in the studies of Doreau et al. (55–92%) [59] and Lock et al. (58–86%) [60]. Furthermore, increasing corn silage proportion in the diets linearly decreased the digestibility of MCFA, LCFA, SFA, n-3 PUFA, and total FA in the small intestine due to the higher duodenal pH [14] and passage rate (Kp) [10,15], and lower concentration of bile acids [10] in the duodenum. Additionally, the hydrolysis and digestion of forage lipids are low, as the lipids are mainly composed of glycolipids and phospholipids, which are wrapped by cell walls [61]. The apparent digestibility of C18:3 n-3 was extremely low (55.67%) in the cows fed CS; the reduction was probably because of the low content of C18:3 n-3 in corn silage. Meanwhile, higher dietary corn silage proportion linearly decreased the apparent digestion of C16:0, total CLA, and n-3 PUFA, and tended to linearly increase the apparent intestinal digestion of total FA and OCFA. The differences between apparent intestinal digestion and digestibility data are because of the differences in FA intake and duodenal FA flow. There are no studies on dairy cows about the molecular digestion of FA from the intestine; thus, further studies are needed to understand the molecular mechanism of FA digestion.

VLDL and chylomicron, which are predominantly comprised of triacylglycerol, cholesteryl ester (CE), and phospholipid fractions, are the major routes for the uptake of dietary LCFA into milk [10]. Although the plasma lipid classes and mammary plasma flow were not measured in this study, the arteriovenous differences in esterified total plasma FA could also partially represent the triacylglycerol uptake by the MG because plasma CE and phospholipid fractions were basically unavailable for mammary uptake [62,63]. In agreement with the pairwise levels of apparent intestinal digestion and milk content, linearly higher arteriovenous differences in C16:0, C18:3 n-3, total CLA, and n-3 PUFA were observed when dietary grass hay proportion increased. With respect to CLA, previous studies have confirmed that endogenous synthesis from trans-11 vaccenic acid by Δ^9^-desaturase in the MG contributes to over 80% of total milk CLA content [64,65]. Increasing the proportion of grass hay in the diets increased the Δ^9^-desaturase activity in the MG, together with the higher apparent intestinal digestion and uptake by the MG of CLA, eventually resulting in higher CLA content (+35.3%) in milk. In response to higher apparent digestion of n-3 PUFA in higher GH, higher arteriovenous differences in milk content (+27.0%) of n-3 PUFA were observed. As consequence, grass hay increased apparent intestinal digestion and uptake by MG of CLA and n-3 PUFA, eventually resulting in higher CLA and n-3 PUFA contents in milk. Although dietary PUFA are preferentially incorporated into plasma CE and phospholipid fractions rather than the triacylglycerol fraction and are subsequently unavailable for mammary uptake, higher proportion of grass hay led to higher contents of n-3 PUFA in milk [19,27,29]. Therefore, further investigations that include the metabolome and functional mechanisms involved in FA uptake and milk fat synthesis in MGs are required.

## 5. Conclusions

Increasing dietary corn silage proportion increased DMI, intestinal digestion of LCFA, arteriovenous differences in VLDL and LCFA, together with a higher content of LCFA in milk, indicating that a higher proportion of corn silage stimulated the preformed synthesis pathways. Conversely, the grass hay diet increased SCFA, total CLA, and n-3 PUFA contents in the milk of cows compared with those fed the corn silage diet. The decrease in SCFA in the milk is predominantly through decreases in acetate production during ruminal fermentation, uptake by MGs, and de novo synthesis pathways when cows were fed the corn silage diet. Increasing the proportion of grass hay in the diets increased intestinal digestion and arteriovenous differences in n-3 PUFA and total CLA, and, together with higher Δ^9^-desaturase activity in the MG, resulted in higher relative contents of n-3 PUFA and total CLA in the milk. This study can enhance our understanding of the overall mechanism of forages regulating HPFA status in milk, and provides potential solutions to improve the utilization of corn silage in dairy diets and their milk quality.

## Figures and Tables

**Figure 1 foods-12-00303-f001:**
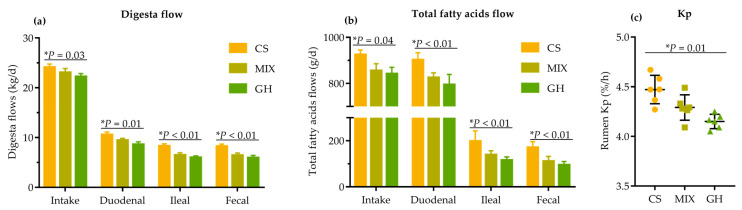
Effects of different dietary forage sources on (**a**) digesta flow (DM) across the GIT; (**b**) total fatty acids flow across the GIT; (**c**) Kp through the rumen of cows. The bars indicate the standard deviation, * represents the linear effect of corn silage proportion in diets.

**Figure 2 foods-12-00303-f002:**
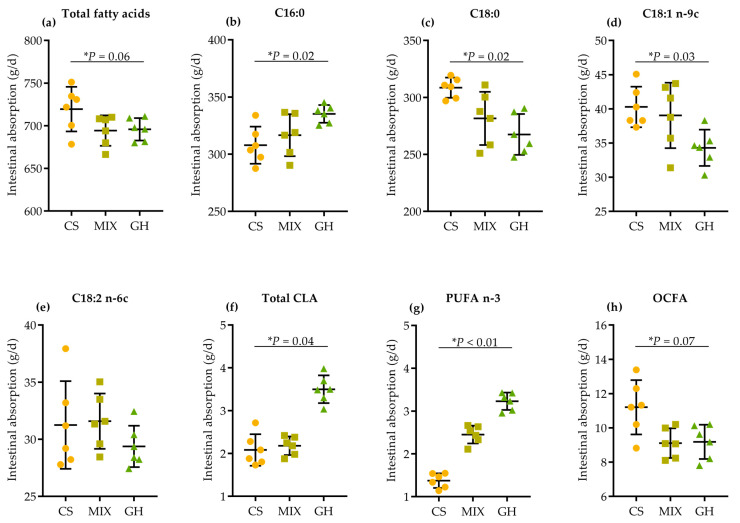
Effects of different dietary forage sources on apparent intestinal absorption of (**a**) total FA, (**b**) C16:0, (**c**) C18:0, (**d**) C18:1 n-9c, (**e**) C18:2 n-6c, (**f**) total CLA, (**g**) n-3 PUFA, and (**h**) OCFA of cows.

**Figure 3 foods-12-00303-f003:**
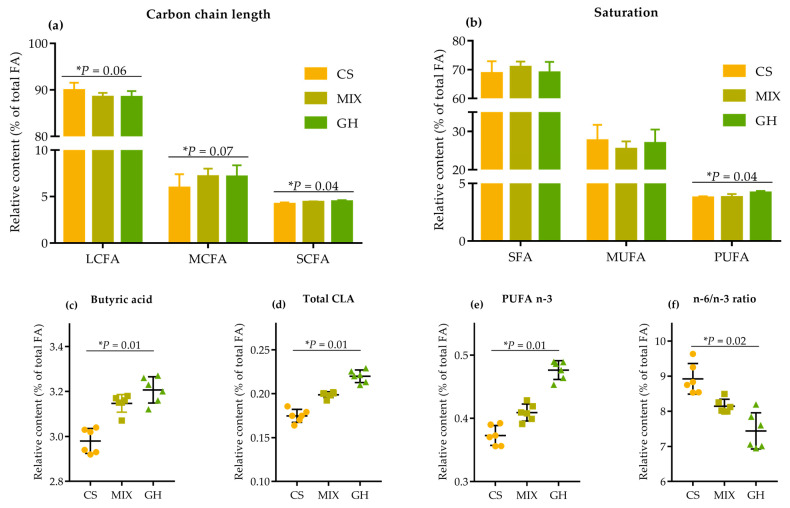
Effects of different forage sources on relative contents of (**a**) carbon chain length of fatty acids, (**b**) saturation of fatty acids, (**c**) butyric acids (C4:0), (**d**) total CLA, (**e**) n-3 PUFA, and (**f**) OCFA in milk. SCFA = short-chain fatty acid: C4:0 + C6:0; MCFA = medium-chain fatty acid: C10:0 + C12:0; LCFA = long-chain fatty acid: carbon chain > 12 (C13:0 to C24:1); SFA = saturated fatty acid: C10:0 + C12:0 + C13:0 + C14:0 + C15:0 + C16:0 + C17:0 + C18:0 + C20:0 + C22:0 + C24:0; MUFA = monounsaturated fatty acid: C16:1 + C18:1 n-9c + C20:1 + C22:1 n-9 + C24:1; PUFA = polyunsaturated fatty acid: C18:3 n-3 + C20:5 n-3 + C18:2 n-6c + C20:3 n-6 + C20:4 n-6; Total CLA= total conjugated linoleic acid: CLA-c9t11 + CLA-t10c12; OCFA = odd-chain fatty acid: C13:0 + C15:0 + C17:0; HFA = hypercholesterolemic fatty acid: C12:0 + C14:0 + C16:0. n-3 PUFA = omega-3 polyunsaturated fatty acid: C18:3 n-3 + C20:5 n-3; n-6 PUFA = omega-6 polyunsaturated fatty acid: C18:2 n-6c + C20:3 n-6 + C20:4 n-6. The bars indicate the standard deviation, * represents the linear effect of corn silage proportions in diets.

**Figure 4 foods-12-00303-f004:**
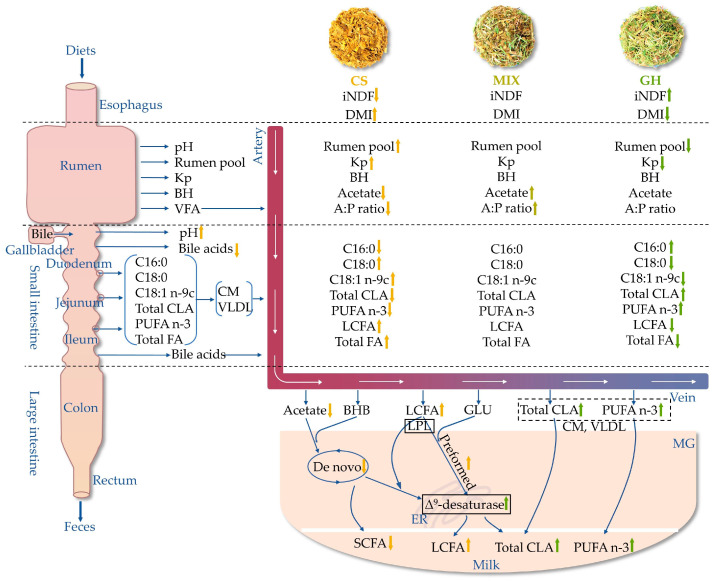
Overall regulatory mechanism of different dietary forage sources regulating health-promoting fatty acid (HPFA) status in milk of lactating cow. iNDF = indigestible neutral detergent fiber, Kp = passage rate out flowing rumen, BH = ruminal biohydrogenation, VFA = volatile fatty acid, A:P ratio = acetate–propionate ratio, Total CLA = total conjugated linoleic acid, n-3 PUFA = omega-3 polyunsaturated fatty acid, BHB = β-hydroxybutyric acid, CM = chylomicron, VLDL = very-low-density lipoprotein, GLU = glucose, MG = mammary gland, ER = endoplasmic reticulum, LPL = lipoprotein lipase. Enzyme and enzyme activity in the black box denote a key role in HPFA metabolism by the MG. Total CLA and n-3 PUFA in the dotted box indicate that they are taken up by the MG in the form of CM or VLDL. The arrows indicate the linear or quadratic effects of corn silage proportions on FA metabolism among the three groups, upward and downward arrows denote positive and negative effects (*p* < 0.10), respectively.

**Table 1 foods-12-00303-t001:** Ingredients, chemical composition (% of DM unless otherwise noted), and fatty acid profile (% of total FA) of the experimental diets.

Item	Diet ^1^		
	CS	MIX	GH
Ingredient			
Corn silage	46.00	23.00	
Alfalfa hay		6.00	12.00
Oat hay		8.00	16.00
Ground corn	14.22	23.90	34.15
Soybean meal	10.37	9.69	8.44
Menergy C ^2^	1.27	1.27	1.27
Concentrate supplement ^3^	23.88	23.88	23.88
Mineral–vitamin premix ^4^	4.26	4.26	4.26
Chemical composition			
Crude Protein (CP)	17.01	17.08	17.04
Neutral Detergent Fiber (NDF)	31.34	29.58	27.94
Indigestible NDF (iNDF_288_) ^5^, % of DM	8.07	8.36	8.34
Indigestible NDF (iNDF_288_) ^5^, % of NDF	25.74	28.25	29.83
Acid Detergent Fiber (ADF)	16.96	16.83	16.79
Starch	25.63	25.84	25.67
Ether Extract	4.41	4.34	4.23
Total fatty acids	3.84	3.71	3.79
NE_L_, MJ/kg of DM	7.20	7.24	7.24
Total fatty acids, % of DM	3.84	3.71	3.79
Fatty acid composition ^6^			
Palmitic acid (C16:0)	46.18	48.27	51.55
Stearic acid (C18:0)	2.88	3.31	3.84
Oleic acid (C18:1 n-9c)	15.70	14.87	13.75
Linoleic acid (C18:2 n-6c)	29.75	27.58	23.99
Linolenic acid (C18:3 n-3)	1.84	2.44	2.58
MCFA	0.62	0.73	1.23
LCFA	99.38	99.27	98.77
SFA	52.07	54.36	59.07
MUFA	16.34	15.63	14.36
PUFA	31.59	30.02	26.56
n-6/n-3 ratio	16.15	11.31	9.30

^1^ CS = diet with 46% corn silage as the sole dietary forage source; MIX = diet with a mixture of 23% corn silage and 14% grass hays (6% alfalfa hay and 8% oat hay) as the dietary forage sources; GH = diet with 28% grass hays (12% alfalfa hay and 16% oat hay) as the sole forage source. ^2^ Menergy C, a commercial palm oil FA product purchased from Wilmar Oleo (Tianjin) Co., Ltd., Tianjin, China, iodine value 8.4 g/100 g, acid value 168 mgKOH/g. ^3^ Concentrate supplement contained (DM basis) 28.18% canola meal, 24.62% cottonseed, 21.19% wheat bran, 6.53% sunflower seed meal, 6.41% beet pulp pellet, 4.86% soybean hull, and 8.21% yeast culture. ^4^ Mineral–vitamin premix, a commercial premix produced by Beijing Sunlon Livestock Development Co. LTD, stated to contain (g/kg of DM): salt (100.043), bicarbonate (203.133), Ca (141.286), Mg (53.686), P (24.014), K (74.157), S (18.104), Co (7. 033), Cu (5.641), Zn (0.809), Mn (0.666), Fe (0.229), I (0.01), Se (0.01); vitamin A (182.835 kIU/kg), vitamin D (30476.385 IU/kg), and vitamin E (558.649 IU/kg). ^5^ iNDF_288_, = indigestible NDF determined by ruminal incubation for 288 h. ^6^ MCFA = medium-chain fatty acids: C10:0 + C12:0; LCFA = long-chain fatty acids: carbon chain > 12 (C13:0 to C24:1); SFA = saturated fatty acids: C10:0 + C12:0 + C13:0 + C14:0 + C15:0 + C16:0 + C17:0 + C18:0 + C20:0 + C22:0 + C24:0; MUFA = monounsaturated fatty acids: C16:1 + C18:1 n-9c + C20:1; PUFA = polyunsaturated fatty acids: C18:2 n-6c + C18:3 n-3; n-6/n-3 ratio = n-6 PUFA / n-3 PUFA ( n-3 PUFA = C18:3 n-3, n-6 PUFA = C18:2 n-6c).

**Table 2 foods-12-00303-t002:** The chemical composition (% of DM unless otherwise noted) and fatty acid profile (% of total FA) of the varied ingredients.

Item	Ingredient					
	Corn Silage	Alfalfa Hay	Oat Hay	Ground Corn	Soybean Meal	Menergy C
Dry matter, % as fed	31.62	91.12	90.10	86.30	87.65	98.12
Neutral Detergent Fiber (NDF)	42.07	44.01	55.46	8.81	9.64	
Indigestible NDF (iNDF288), % of DM	11.19	20.83	20.99			
Indigestible NDF (iNDF288), % of NDF	26.59	47.34	37.84			
Acid Detergent Fiber (ADF)	24.62	31.61	32.09	2.45	5.32	
Ether Extract	3.49	2.19	2.08	4.59	2.43	99.58
Total fatty acids, % of DM	2.39	1.49	0.86	4.38	2.23	97.93
Fatty acid composition						
Palmitic acid (C16:0)	19.73	31.79	28.60	16.24	16.51	85.21
Stearic acid (C18:0)	2.65	5.14	3.58	1.55	4.70	4.77
Oleic acid (C18:1 n-9c)	18.96	8.83	11.56	23.07	13.78	4.92
Linoleic acid (C18:2 n-6c)	43.16	21.94	22.29	55.84	51.96	3.03
Linolenic acid (C18:3 n-3)	10.63	25.14	22.49	1.53	9.30	0.09
MCFA	0.44	0.42	1.12	0.13	1.69	0.09
LCFA	99.56	99.58	98.88	99.87	98.31	99.91
SFA	26.39	42.82	42.44	19.02	24.62	91.85
MUFA	19.75	10.09	12.78	23.56	14.09	5.02
PUFA	53.86	47.09	44.78	57.43	61.30	3.13
n-6/n-3 ratio	4.06	0.87	0.99	36.42	5.59	32.54

**Table 3 foods-12-00303-t003:** Effects of different forage sources on dry matter intake (DMI) and lactation performance of cows.

Item	Diet			SEM	Probability ^1^		
	CS	MIX	GH		T	L	Q
DMI, kg/d	24.11	23.09	22.24	0.84	0.04	0.03	0.50
Milk yield, kg/d	26.16	25.94	26.92	2.21	0.69	0.73	0.38
3.5% FCM ^2^, kg/d	30.77	29.70	31.26	2.43	0.58	0.67	0.19
ECM ^3^, kg/d	31.22	30.37	31.67	2.47	0.53	0.64	0.18
Feed efficiency ^4^	1.27	1.32	1.41	0.02	0.03	0.08	0.58
Milk fat, %	4.54	4.38	4.49	0.18	0.62	0.56	0.23
Milk fat yield, kg/d	1.20	1.14	1.21	0.04	0.58	0.71	0.16
Milk protein, %	3.78	3.80	3.72	0.05	0.19	0.74	0.49
Milk protein yield, kg/d	0.99	0.99	1.00	0.05	0.83	0.19	0.73
Milk lactose, %	5.08	5.02	5.12	0.06	0.20	0.65	0.84
Milk lactose yield, kg/d	1.33	1.30	1.38	0.22	0.62	0.41	0.68

^1^ Probability of treatment effects: T = fixed effect of diet treatment; L = linear effect of corn silage proportions in diets; Q = quadratic effect of corn silage proportions in diets. ^2^ 3.5% fat-corrected milk (3.5% FCM) (kg/d) = (0.4324 × kg of milk yield) + (16.216 × kg of milk fat). ^3^ Energy-corrected milk (ECM) (kg/d) = (0.327 × kg of milk yield) + (12.95 × kg of milk fat) + (7.2 × kg of milk protein). ^4^ Feed efficiency = 3.5% FCM/DMI.

**Table 4 foods-12-00303-t004:** Effects of different forage sources on the ruminal and duodenal fermentation characteristics of cows.

Item	Diet			SEM	Probability		
	CS	MIX	GH		T	L	Q
Fermentation characteristics in rumen							
pH	6.47	6.46	6.55	0.72	0.89	0.72	0.81
Total VFA, mM	116.78	120.85	119.56	1.39	0.51	0.62	0.31
Acetate, mM	67.21	70.80	69.05	0.81	0.06	0.98	0.04
Propionate, mM	29.34	27.43	28.30	0.71	0.30	0.61	0.15
Butyrate, mM	13.43	15.15	14.69	0.39	0.29	0.42	0.37
Isobutyrate, mM	0.91	0.78	1.19	0.12	0.56	0.22	0.05
A:P ratio ^1^	2.30	2.58	2.44	0.06	0.02	0.59	0.02
Fermentation characteristics in duodenum							
pH	4.12	4.01	3.87	0.11	0.08	0.07	0.96
Total VFA, mM	10.11	11.49	10.04	0.49	0.09	0.81	0.08
Acetate, mM	7.01	7.80	6.81	0.37	0.13	0.73	0.26
Propionate, mM	2.95	3.51	3.09	0.15	0.04	0.89	0.52
Butyrate, mM	0.15	0.18	0.14	0.01	0.60	0.67	0.37
A:P ratio ^1^	2.35	2.25	2.21	0.07	0.66	0.49	0.86
R–D of acetate ^2^, mM	60.20	63.02	62.24	0.69	0.05	0.98	0.03
R–D of butyrate ^2^, mM	13.28	14.97	14.55	0.33	0.26	0.40	0.31

^1^ A:P ratio = ratio of acetate to propionate. ^2^ R–D differences in VFA = molar concentration in rumen—molar concentration in duodenum.

**Table 5 foods-12-00303-t005:** Effects of different dietary forage sources on bile acid profile (ng/g of DM) in duodenal and ileal digesta of cows.

Item	Diet			SEM	Probability		
	CS	MIX	GH		T	L	Q
Bile acid profile in duodenal digesta							
Total bile acids, ×10^7^	10.61	13.53	12.27	0.58	0.09	0.29	0.06
Lithocholic acid (LCA), ×10^4^	0.79	1.81	1.47	0.17	0.01	0.04	0.01
Hyodeoxycholic acid (HDCA), ×10^4^	2.47	7.74	3.13	0.91	0.01	0.86	<0.01
Chenodeoxycholic acid (CDCA), ×10^5^	0.61	3.11	2.72	0.43	<0.01	0.02	0.01
Deoxycholic acid (DCA), ×10^5^	5.60	9.70	10.44	0.93	0.04	0.03	0.13
β-muricholic acid (β-MCA), ×10^3^	3.84	8.10	5.05	0.65	<0.01	<0.01	<0.01
Cholic acid (CA), ×10^6^	2.16	8.70	8.08	1.10	<0.01	<0.01	<0.01
Glycolithocholic acid (GLCA), ×10^4^	2.06	4.54	2.85	0.38	<0.01	0.17	<0.01
Glycoursodeoxycholic acid (GUDCA), ×10^3^	4.65	3.90	3.81	0.78	0.92	0.74	0.83
Glycochenodeoxycholic acid (GCDCA), ×10^6^	1.07	1.09	0.92	0.09	0.79	0.56	0.77
Glycodeoxycholic acid (GDCA), ×10^6^	2.75	4.23	2.73	0.35	0.11	0.69	0.04
Glycocholic acid (GCA), ×10^7^	1.48	1.75	1.80	0.10	0.39	0.25	0.50
Taurolithocholic acid (TLCA), ×10^6^	2.83	3.29	2.80	0.18	0.53	0.82	0.29
Taurochenodeoxycholic acid (TCDCA), ×10^6^	2.44	6.40	2.18	0.84	0.04	0.52	0.02
Taurodeoxycholic acid (TDCA), ×10^6^	5.87	8.71	3.73	0.86	0.03	0.08	0.02
Taurocholic acid (TCA), ×10^7^	7.34	8.40	8.29	0.37	0.51	0.09	0.12
Bile acid profile in ileal digesta							
Total bile acids, ×10^5^	7.22	5.90	4.79	0.54	0.19	0.08	0.69
Lithocholic acid (LCA), ×10^3^	4.66	5.60	3.29	0.47	0.12	0.13	0.13
Hyodeoxycholic acid (HDCA), ×10^4^	1.24	1.92	0.79	0.22	0.10	0.22	0.07
Chenodeoxycholic acid (CDCA), ×10^4^	4.69	6.90	2.80	0.81	0.10	0.17	0.08
Deoxycholic acid (DCA), ×10^5^	1.69	2.79	0.75	0.36	0.04	0.09	0.03
Cholic acid (CA), ×10^5^	7.21	10.71	2.39	1.40	0.02	0.03	0.02
Glycolithocholic acid (GLCA), ×10^3^	3.10	5.31	2.87	0.66	0.28	0.69	0.13
Glycochenodeoxycholic acid (GCDCA), ×10^4^	2.61	2.65	2.41	0.29	0.95	0.80	0.87
Glycodeoxycholic acid (GDCA), ×10^4^	8.25	6.74	5.43	1.01	0.59	0.33	0.85
Glycocholic acid (GCA), ×10^5^	1.98	1.67	1.56	0.21	0.75	0.50	0.75
Taurolithocholic acid (TLCA), ×10^3^	6.43	9.04	4.79	1.18	0.38	0.47	0.24
Taurochenodeoxycholic acid (TCDCA), ×10^4^	2.74	3.63	1.99	0.58	0.58	0.55	0.41
Taurodeoxycholic acid (TDCA), ×10^4^	3.40	2.65	2.39	0.35	0.54	0.33	0.64
Taurocholic acid (TCA), ×10^5^	4.56	3.51	2.74	0.36	0.10	0.04	0.56

**Table 6 foods-12-00303-t006:** Effects of different dietary forage sources on arteriovenous (AV) differences in dominant and grouped fatty acids (mg/100 mL) and other metabolites (mM unless otherwise noted) in the mammary glands of cows.

Item	Diet			SEM	Probability		
	CS	MIX	GH		T	L	Q
AV difference in dominant and grouped fatty acids ^1^							
C16:0	6.80	7.46	7.13	0.86	0.09	0.06	0.26
C18:0	10.25	11.30	8.32	1.27	0.48	0.34	0.15
C18:1 n-9c	4.42	3.48	4.30	1.02	0.02	0.90	0.08
C18:2 n-6c	26.20	18.92	21.45	1.82	0.26	0.30	0.12
CLA-c9t11	0.20	0.31	3.38	0.03	0.02	0.04	0.41
C18:3 n-3	0.92	1.57	1.81	0.43	0.07	0.04	0.09
MCFA	0.07	−0.02 ^2^	−0.01 ^2^	0.03	0.37	0.27	0.41
LCFA	55.08	47.84	49.35	3.24	0.08	0.19	0.09
SFA	19.95	21.15	17.40	1.43	0.58	0.52	0.47
MUFA	3.44	5.69	5.66	0.93	0.03	0.85	0.30
PUFA	30.03	24.03	26.77	1.90	0.56	0.97	0.33
Total CLA	0.21	0.33	0.39	0.03	0.02	0.04	0.42
OCFA	1.11	1.74	0.53	0.25	0.31	0.49	0.10
HFA	6.89	7.43	7.43	0.74	0.60	0.55	0.94
n-3 PUFA	0.89	1.08	2.18	0.41	0.03	0.04	0.14
n-6 PUFA	28.97	24.59	25.88	1.85	0.49	0.41	0.52
Total fatty acids	55.15	47.83	49.34	3.24	0.08	0.19	0.09
AV difference in other metabolites ^3^							
Acetate	0.94	1.16	1.11	0.04	0.08	0.01	0.01
BHBA	0.28	0.29	0.28	0.02	0.98	0.95	0.83
VLDL	0.81	0.68	0.73	0.13	0.81	0.13	0.07
GLU	0.75	0.75	0.76	0.12	0.95	0.99	0.98
NEFA, umol/L	−29.80 ^2^	−32.65 ^2^	−30.19 ^2^	1.51	0.80	0.94	0.48

^1^ MCFA = medium-chain fatty acid: C10:0 + C12:0; LCFA = long-chain fatty acid: carbon chain > 12 (C13:0 to C24:1); SFA = saturated fatty acid: C10:0 + C12:0 + C13:0 + C14:0 + C15:0 + C16:0 + C17:0 + C18:0 + C20:0 + C22:0 + C24:0; MUFA = monounsaturated fatty acid: C16:1 + C18:1 n-9c + C20:1 + C22:1 n-9 + C24:1; PUFA = polyunsaturated fatty acid: C18:3 n-3 + C20:5 n-3 + C18:2 n-6c + C20:3 n-6 + C20:4 n-6; Total CLA= total conjugated linoleic acid: CLA-c9t11 + CLA-t10c12; OCFA = odd-chain fatty acid: C13:0 + C15:0 + C17:0; HFA = hypercholesterolemic fatty acid: C12:0 + C14:0 + C16:0. n-3 PUFA = omega-3 polyunsaturated fatty acid: C18:3 n-3 + C20:5 n-3; n-6 PUFA = omega-6 polyunsaturated fatty acid: C18:2 n-6c + C20:3 n-6 + C20:4 n-6. ^2^ Negative values indicate a net efflux of FAs from the MG into blood. ^3^ BHB = β-hydroxybutyric acid, VLDL = very-low-density lipoprotein, GLU = glucose, NEFA = nonesterified fatty acid.

**Table 7 foods-12-00303-t007:** Effects of different dietary forage sources on synthesized fatty acid origins (% of total fatty acids) and desaturase activity in milk of cows.

Item	Diet			SEM	Probability		
	CS	MIX	GH		T	L	Q
Synthesized FA origin ^1^							
De novo	21.48	24.80	23.83	0.81	0.02	0.20	0.15
Mixed	39.24	40.38	39.27	0.60	0.69	0.48	0.69
Preformed	39.28	34.82	36.90	1.00	0.06	0.61	0.07
Desaturase activity							
C14:1/(C14:0 + C14:1) ^2^	0.086	0.092	0.095	0.003	0.02	0.06	0.89
C16:1/(C16:0 + C16:1) ^2^	0.051	0.057	0.060	0.003	0.05	0.03	0.92
C18:1/(C18:0 + C18:1) ^2^	0.70	0.71	0.72	0.01	0.06	0.08	0.85
C20:1/(C20:1 + C20:0) ^2^	0.36	0.36	0.36	0.01	0.91	0.92	0.70
C20:4 n-6/(C20:4 n-6 + C20:3 n-6) ^3^	0.48	0.51	0.50	0.02	0.15	0.75	0.58
C20:3 n-6/(C20:3 n-6 + C20:2 n-6) ^4^	0.84	0.87	0.84	0.01	0.34	0.81	0.24

^1^ De novo = C4 to C15, mixed = C16:0 + C16:1 + C17:0, preformed = C18:0 and longer. ^2^ Δ^9^-Desaturase activity = milk FA product/(product + substrate), indicates that stearoyl-CoA desaturase catalyzes the introduction of a cis double bond at Δ^9^. ^3^ Δ^5^-Desaturase activity = milk FA product/(product + substrate), indicates that FA desaturase 1 catalyzes the introduction of a cis double bond at Δ^5^. ^4^ Δ^8^-Desaturase activity = milk FA product/(product + substrate), indicates that FA desaturase 2 catalyzes the introduction of a cis double bond at Δ^8^.

## Data Availability

Data are contained within the article and Appendix A.

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
