# Peer review of "Feeding Corn Silage or Grass Hay as Sole Dietary Forage Sources: Overall Mechanism of Forages Regulating Health-Promoting Fatty Acid Status in Milk of Dairy Cows"

_foods, 2023, doi:10.3390/foods12020303_

Round 1

Reviewer 1 Report

Dear authors,

I reviewed your manuscript identified as foods-2122286 (Feeding corn silage or grass hay as sole dietary forage sources: overall mechanism of forage regulating health-promoting fatty acids in milk of dairy cows). In my opinion, your research is very interesting and can effectively contribute to broadening the knowledge of the mechanisms involved in the improvement of the fatty acids profile of bovine milk through dietary manipulation. Thus, I positively evaluated the manuscript, both for scientific robustness and for structure and form of presentation point of view. However, apart from a few minor details which you will find listed below, I have some perplexity. First, I would propose making the discussions one paragraph, but above all, I ask you to make a paragraph on conclusions. The results presented are multiple and relevant, so I believe that the conclusions can help the reader focus on the main results achieved by the study. Simply for this reason I requested a major revision.

As a minor detail, I suggest to:

-          add to the notes in table 1 (L 132) the equation used to estimate the energy content of diets

-          indicate the presence of a Roughage Intake Control System (L 153) on the description of the pen (L 115-116)

-          remove the underlining from references along the text

-          update the statement at lines 451-452 with the following reference https://doi.org/10.3168/jds.2019-17155

Author Response

Point 1: First, I would propose making the discussions one paragraph, but above all, I ask you to make a paragraph on conclusions. The results presented are multiple and relevant, so I believe that the conclusions can help the reader focus on the main results achieved by the study. Simply for this reason I requested a major revision.

Response 1: Thank you for the comments. We have modified the discussion and made a paragraph on conclusions according to your suggestion. Please check the revised manuscript. (Page 15-16, Line 494-521; Page 18-19, Line 667-791)

minor details:

Point 2: add to the notes in table 1 (L 132) the equation used to estimate the energy content of diets

Response 2: Thank you for the comment. As suggested, we have added the description and cited reference in “2.4. Analytical procedures” as follows: “Concentration of NEL of the diets was calculated according to NRC (2001) [31].” (Page 7, Line 256-257)

Point 3: indicate the presence of a Roughage Intake Control System (L 153) on the description of the pen (L 115-116)

Response 3: Thank you for the comment. We have rewritten the sentence to state clearly according to your suggestion. “Cows were housed in an individual pen equipped with Roughage Intake Control System (RFID, Zhenghong Company, Shanghai, China). The pen was bedded with rice hull that was cleaned every 3 d by removing all bedding and renewing it with fresh rice hull.” (Page 3, Line 126-127)

Point 4: remove the underlining from references along the text

Response 4: Thank you for the comment. As suggested, we have removed the underlining from references along the text.

Point 5: update the statement at lines 451-452 with the following reference https://doi.org/10.3168/jds.2019-17155

Response 5: Thank you for the comment. Reviewer 2 suggested that we should provide the key findings in the first paragraph of discussion. So, we deleted the paragraph and updated the statement with your suggested reference. (Page 2, Line 83-84)

Reviewer 2 Report

The study describe potential effect of forages on milk quality. This study highlighted effect of forages and their dietary FA composition, DMI, ruminal fermentation and biohydrogenation, apparent intestinal digestion, arteriovenous differences, and desaturase activity in the MG in lactating cows. It provides an insight for specifically for digestibility but detailed profile of milk is needed to establish a link as regulation of forages impact. Overall study is well-designed and fulfill the criteria to make space in the journal. There are some specific comments which can be seen in attached file.

Author Response

The study describes potential effect of forages on milk quality. This study highlighted effect of forages and their dietary FA composition, DMI, ruminal fermentation and biohydrogenation, apparent intestinal digestion, arteriovenous differences, and desaturase activity in the MG in lactating cows. It provides an insight for specifically for digestibility but detailed profile of milk is needed to establish a link as regulation of forages impact. Overall study is well-designed and fulfill the criteria to make space in the journal. There are some specific comments which can be seen in attached file.

Point 1: It provides an insight for specifically for digestibility but detailed profile of milk is needed to establish a link as regulation of forages impact.

Response 1: Thank you very much for your comments. We have provided the FA profile of milk in the supplementary materials (Table 8).

some specific comments:

Point 2: Can be modify the title as (Health-promoting fatty acids status in milk of lactating cows fed through corn silage or hay as sole dietary forages). Because it is based on observation and authors did not observe the cellular changes or molecular aspects of systems. (Page 1, Line 2- 4)

Response 2: Thank you very much for the comments. Actually, the title has also puzzled us for a long time, and we have rewritten the title several times to state clearly. As you mentioned, we did not observe the cellular changes or molecular aspects of systems, so it seems unreasonable to use “mechanism” in the title. Although the fatty acid composition of milk depends on lipid metabolism across several organs (such as liver, GIT, and MG), it is mainly achieved by influencing the digestion and absorption of fatty acids in the GIT. And the dietary iNDF contents regulate the DMI and digesta flows across the GIT. This study aimed to determine the effect of different dietary forage sources on dietary iNDF content and fatty acid composition, DMI, ruminal fermentation and biohydrogenation, apparent intestinal digestion, arteriovenous differences, and desaturase activity in the MG in lactating cows. Finally, the overall mechanism of different dietary forage sources regulating the HPFA in milk of lactating cows was summarized based on the present results. Together with your comments, all the authors agreed to modify title to “Feeding corn silage or grass hay as sole dietary forage sources: overall mechanism of forages regulating health-promoting fatty acids status in milk of dairy cows”. Of course, if you insist on modifying the title as Health-promoting fatty acids status in milk of lactating cows fed through corn silage or hay as sole dietary forages, I will modify it as your recommendation.

Point 3: Need grammatical correction for better expressions. Precise methodology should be included. Conclusion is not reflecting the essence of study, rather than statement. (Page 1, Line 15- 28)

Response 3: Thank you very much for your comments and sorry for the inaccurate description for the methodology and conclusion. Foods requires a single paragraph of about 200 words maximum for the abstract and our results were multiple and relevant, so it was quite difficult to write the abstract precisely. We tried to summarize the effects of different dietary forage sources on the de novo and preformed synthesis pathways. We have modified the abstract please see the revised manuscript. (Page 1, Line 15- 29)

Point 4: This section provides bit better insight about feeding mechanism based previous studies. Last para should be modified in view of study objectives. Additionally, English language proofreading will improve the readability. (Page 1-2, Line 33- 95)

Response 4: Thank you very much for your comments. As suggested, we have rewritten the last paragraph in view of study objectives. Please check the revised manuscript. (Page 2, Line 94-99)

Point 5: Did authors observed BW later days?? for diet effects..... (Page 3, Line 115)

Response 5: We did not observe the BW in the later days, because it was a bit difficult and dangerous for the cannulated cows to walk a long way to weigh them. About 10 cm of the T-shaped intestinal cannulas were exposed outside of the body, we were worried that the cows would crash the intestinal cannulas. We checked the body condition score (BCS), peNDF contents, feeding and rumination behaviors, serum hormone and immune levels, and nutrient digestibility, we will publish the data in another paper.

Point 6: Here it is need to present the chemical composition of 03 diets rather than the ingredients details. Merge the Table 1 & 2 for chemical composition of each diet. Actually authors observed the impact of diet not the ingredient effects. (Page 3-4, Line 120-151)

Response 6: Thank you very much for the comments. We have merged the Table 1 & 2 for chemical composition of each diet. We keep the Table 2 as the chemical composition and fatty acid profile of the ingredients. It helps in discussing the effects of ingredients on iNDF contents and FA composition of diets. Please check the revised manuscript. (Page 4, Line 133-151, Table 1; Page 5, Line 166-168, Table 2)

Point 7: Also provide the comparison of feed as advantageous one concerning the chemical composition. (Page 7, Line 303)

Response 7: Thank you very much for the comments. We have provided the comparison of diets according to your suggestion. Please check the revised manuscript. (Page 8, Line 341-346)

Point 8: Present accurately the order of Tables. (Page 8, Line 319)

Response 8: Thank you. Done. (Page 9, Line 360)

Point 9: provide the key findings here in first paragraph. (Page 14, Line 451)

Response 9: Thank you very much for the comments. We have provided the key findings and deleted the first paragraph according to your suggestion. Please check the revised manuscript. (Page 15-16, Line 494-521)

Point 10: The authors cannot claim this because they also observed the responses of systems rather than the biology. (Page 14, Line 450-461)

Response 10: Thank you very much for the comments. We have deleted this paragraph. (Page 15-16, Line 494-521)

Point 11: How modification of existing forages could improve the milk traits? Needs explanation here. What other factors could be influencing in milk quality in studied groups? (Page 15, Line 463-484)

Response 11: Thank you very much for your comments. In this paragraph, we focused on the DMI and lactation performance of dairy cows fed corn silage or grass hay as sole dietary forage sources. They were the most contentious effects of different forage sources on dairy cows. Corn silage quality, such as iNDF content, peNDF content, fermentation parameters, and mycotoxin content, are the limiting factors for high proportion utilization of corn silage in dairy diets. For the milk quality (especially FA composition), we discussed the effects of different forage sources on milk FA composition in the following paragraphs. (Page 15-16, Line 494-521)

Round 2

Reviewer 1 Report

Dear authors,
I have reviewed the revised version of the manuscript identified as foods-2122286. Based on the changes made and, above all, considering the addition of congruous conclusions to the manuscript, I have no doubts about the eligibility of the manuscript for publication. In my opinion, the current version needs no further changes. I congratulate you on your excellent research. Good luck